# The Potential of Aptamer-Mediated Liquid Biopsy for Early Detection of Cancer

**DOI:** 10.3390/ijms22115601

**Published:** 2021-05-25

**Authors:** Dhruvajyoti Roy, Andreas Pascher, Mazen A. Juratli, Judith C. Sporn

**Affiliations:** 1Helio Health, Irvine, CA 92618, USA; 2Department of General, Visceral and Transplant Surgery, University Hospital Münster, 48149 Münster, Germany; andreas.pascher@ukmuenster.de (A.P.); mazen.juratli@ukmuenster.de (M.A.J.); judith.sporn@ukmuenster.de (J.C.S.)

**Keywords:** early cancer detection, circulating biomarkers, liquid biopsy, aptamer, extracellular vesicles, non-invasive diagnosis, circulating tumor cells, DNA-nanostructures

## Abstract

The early detection of cancer favors a greater chance of curative treatment and long-term survival. Exciting new technologies have been developed that can help to catch the disease early. Liquid biopsy is a promising non-invasive tool to detect cancer, even at an early stage, as well as to continuously monitor disease progression and treatment efficacy. Various methods have been implemented to isolate and purify bio-analytes in liquid biopsy specimens. Aptamers are short oligonucleotides consisting of either DNA or RNA that are capable of binding to target molecules with high specificity. Due to their unique properties, they are considered promising recognition ligands for the early detection of cancer by liquid biopsy. A variety of circulating targets have been isolated with high affinity and specificity by facile modification and affinity regulation of the aptamers. In this review, we discuss recent progress in aptamer-mediated liquid biopsy for cancer detection, its associated challenges, and its future potential for clinical applications.

## 1. Introduction

The early detection of cancer increases the chances for a cure and the long-term survival of cancer patients [1,2] and is therefore of the utmost importance for optimized cancer treatment. Currently, most cancers lack effective non-invasive screening tests and are therefore detected too late. Liquid biopsy is a promising tool for the early detection of cancer [3,4], and significant advancement of the technique shows potential as a diagnostic tool for a variety of cancer types, including melanoma [5,6,7,8], breast [9,10,11], colorectal [12,13,14], lung [15], liver [16,17,18], ovarian [19], pancreatic [20], and gastric cancers [21]. The most common analytical targets of liquid biopsy are circulating tumor cells (CTCs), circulating tumor DNA (ctDNA), microRNA, and extracellular vesicles (EVs) [22,23,24,25,26]. Various biological fluids including peripheral blood, urine, pleural fluid, ascites, seminal fluid, and cerebrospinal fluid (CSF) are used to isolate circulating targets for diagnostic applications. However, it is challenging to preserve the biological functions and viability of the analytes isolated from liquid biopsy specimens.

Therefore, there is an urgent need for efficient procedures to specifically and reliably detect targets in the analyte of choice. In particular, the affinity-based capture of analytes from liquid biopsy relies on the specificity and binding affinity of the ligands toward the surface receptors of the analytes. Recently, nucleic acid aptamers have been extensively studied as a powerful detection method. Aptamers are single-stranded oligonucleotides with unique tertiary structures capable of specifically binding to a wide range of targets, including proteins [27], small molecules [28], metal ions [29], viruses [30], bacteria [31], and whole cells [32].

In the recent past, nucleic acid aptamers have emerged as attractive alternatives to antibodies and small molecules in liquid biopsy-based diagnostic, imaging, and targeting applications [33,34,35]. Unlike antibodies, aptamers are easily evolved, synthesized, tailored, and engineered by in vitro methods against the target to find the tightest binding candidates with prominent characteristics for liquid biopsy. In addition, their versatility in structural design and stability greatly the improves sensitivity and high-accuracy large-scale production of aptamers for liquid biopsy applications. In this review, we discuss the recent advances and challenges in the development of aptamer-based liquid biopsy (Figure 1), with a particular focus on the isolation and detection of CTCs, EVs, and other analytes.

## 2. SELEX for Aptamer Discovery

SELEX—systematic evolution of ligands by exponential enrichment—has been developed as a method to select aptamers from liquid biopsy specimens [36]. SELEX was first reported by Tuerk et al. and Ellington et al. in 1990 [27,37]. The general procedure consists of the following steps: (1) the desired selection targets, traditionally proteins, are incubated with a random library consisting of 10^12^ to 10^14^ unique single-stranded oligonucleotides, 20 to 80 bases in length. (2) The target-binding oligonucleotides are separated from the unbound ones. (3) The target-binding oligonucleotides are eluted and amplified by PCR to create the library for the next round of selection. (4) Several rounds of selection are performed with increasing selection pressure to obtain target specific aptamers. (5) The obtained aptamer candidates are sequenced and further characterized (Figure 2).

Existing SELEX methods can be categorized by target types, which include purified proteins, whole cells, and tissues [38,39,40,41]. SELEX methods are onerous and time-consuming procedures, and the selected aptamers might display an unsatisfactory recognition performance despite successful selection. In the following, we will discuss certain modifications that have been implemented to improve selection efficiency and optimize aptamer properties for use with liquid biopsy.

### 2.1. High-Throughput SELEX

The selection efficiency of SELEX has been improved by a set of strategies [42,43]. In order to isolate aptamers from random sequences with high specificity and affinity, it is critical to have a diverse library during the selection process and to minimize technical bias. The multiple cycles of conventional PCR used for traditional SELEX run the risk of amplifying nonspecific byproducts, thus introducing bias during the amplification steps [44]. On one hand, some sequences may be preferentially processed by DNA polymerase and thus be over-enriched during PCR. On the other hand, sequences with complex structures may be more difficult to amplify and therefore be eliminated over the course of the procedure. Novel PCR technologies such as droplet digital [45] and emulsion PCR [46,47,48] have helped to optimize amplification and preserve library diversity by decreasing the accumulation of byproducts and minimizing PCR bias.

After the completion of amplification, the final PCR products are usually cloned into *Escherichia coli* for identification via sequencing. This step is lengthy and tedious, and the resulting clones do not necessarily represent the entire population of aptamers. Inefficient cloning strategies or a small number of clones can miss infrequent aptamers despite excellent affinity. Cloning bias can be bypassed by utilizing high-throughput sequencing and bioinformatics analysis in combination with SELEX (HT-SELEX). HT-SELEX allows for the visualization of dynamic changes among millions of sequence reads throughout the selection process. This reduces cloning bias and enables the identification of high-affinity aptamers much earlier in the selection.

### 2.2. Cell SELEX

Traditional in vitro SELEX uses purified proteins as targets for selecting aptamers. Yet, aptamers binding to purified proteins might not necessarily be able to bind to the same proteins within the cell due to varying protein levels and specific cellular conditions. Cell SELEX, however, which was firstly developed by Daniels et al. [49], uses the whole cell as a target. This increases the chance that the selected aptamers will be able to bind to the target within the physiological context and can be directly applied for diagnostic and therapeutic purposes. Cell SELEX has several advantages over in vitro SELEX: first, targets on the cell surface are presented in their natural conformation and the selected aptamers represent the final results; second, there is no need for time-consuming protein purification— there is actually not even a need for prior identification or characterization of the molecular targets on the cell surface; third, cell SELEX represents an unbiased method for the discovery of new biomarkers and even unknown surface molecules.

Currently, several cell SELEX modifications have been developed to enhance successful aptamer screening. Hicke et al. developed a hybrid system linking the advantages of cell SELEX and traditional SELEX based on purified proteins [50]. In the first step, aptamer candidates were selected against native tenascin-C by incubating the random library with tenascin-C-bearing cells. The following selection rounds were performed using purified tenascin-C protein as the target.

In order to overcome the tedious and time-consuming task of the purification of the recombinant proteins, Ohuchi et al. developed the so-called target expressed on cell surface-SELEX (TECS SELEX) method [51]. TECS SELEX is a SELEX method that directly uses a cell-surface displaying recombinant proteins as the selection target. Using this approach, they successfully identified RNA aptamers binding to TGFBR3 (transforming growth factor-β receptor III) expressed on Chinese hamster ovary cells. Chen et al. used this approach to successfully select aptamers against DC-SIGN (dendritic cell-specific intercellular adhesion molecule-3-grabbing non-integrin) on the surface of NIH3T3 cells [52].

Another recent hybrid method by Soldevilla et al. combines cell SELEX and peptide SELEX. Using this method, they were able to target chemotherapy-resistant tumors expressing MRP1 [53]. Further, cell SELEX-based modifications and hybrid methods include FACS-SELEX [54,55], 3D cell SELEX [56], and cell-internalization SELEX [57,58].

### 2.3. Microfluidic SELEX

In 2006, Hybarger et al. combined traditional SELEX with a microfluidic system introducing microfluidic SELEX or M-SELEX [59]. The prototype consists of microlines that are loaded with reagents, a pressurized reagent reservoir and distributor, a PCR cycler, and operable valves for selection and sample routing. Using this prototype, they successfully selected an RNA aptamer-binding lysozyme. In 2009, Luo et al. developed an automated, more rapid aptamer selection system [60]. For this approach, a magnetic bead-based SELEX is combined with microfluidic technology, and a continuous-flow magnetically activated chip is used as a separation device. Following this strategy, the enriched aptamer pool, which was obtained after only a single round of selection, was able to bind to recombinant botulinum neurotoxin type A with high affinity. However, several technical challenges were encountered. Aggregation of the magnetic beads in the microchannels influenced the purity and yield of the recovery. In addition, the flow streams could be distorted by microbubbles in the microchannels. To overcome these challenges, Soh and coworkers further advanced the M-SELEX method by furnishing the microchannel with magnetic materials, thus successfully isolating aptamers targeting streptavidin with a Kd value of 25 nM in only three selection rounds [60].

Another novel microfluidic SELEX, developed by Park et al. in 2009, incorporates nanoporous sol-gel material from protein microarrays [61]. Sol-gels can hold a large number of target molecules due to their nanoporous structure, which facilitates the selection of aptamers against multiple targets. Several target molecules with Kd values in the low nanomolar range have been used with sol-gel SELEX. Using this approach, the efficiency of aptamer selection is improved, and only five to eight rounds of selection are usually needed [62,63,64]. However, some of the concerns remain in sol-gel SELEX, such as molecule integrity and stability throughout the multiple selection cycles.

Other modifications of microfluidic techniques aiming at enhancing the efficiency of selecting aptamers include capillary electrophoresis (CE) microfluidic SELEX [65], bead-based microfluidic SELEX [66,67], and protein microarray-microfluidic chip SELEX [68].

## 3. Development of Cancer-Specific Aptamers for Diagnosis and Treatment

The majority of traditional cancer drugs are neither specific nor selective and can cause serious side effects and toxicity within the body during and after treatment [69,70]. The use of aptamers in cancer treatment opens an exciting new avenue. Aptamers are a class of small nucleic acid ligands with high affinity and specificity for their targets [27,37]. Such targets can be either cancer-specific molecules or biomarkers that are related to the development of cancer. Aptamers selected against these targets can be used as drugs themselves or as vehicles to deliver drugs to a certain target after coupling the aptamers with drugs, siRNA, nanoparticles, etc. This creates a powerful targeted drug delivery system that is specific for the selected target and minimizes toxicity to normal cells. This can potentially reduce the dose required for treatment and, thus, enhances therapeutic efficacy.

Thus far, a number of aptamers that target tumor cells have been identified by SELEX: A10, an anti-prostate-specific membrane antigen (PSMA) aptamer [71]; AS1411, an anti-nucleolin aptamer [72,73]; EpCAM, anti-epithelial cell adhesion molecule aptamer [74,75]; Sgc8, an anti-protein tyrosine kinase 7 (PTK7) aptamer [76,77,78] and MUC1, anti-mucin1 aptamer [79]. In addition, a variety of drug delivery systems have been developed using the aptamers for the targeted treatment of specific cancer cells.

Early and exact cancer diagnosis is of great clinical significance as it can help doctors to implement the best treatment strategies in a timely manner, evaluate treatment response, and monitor for recurrence or metastasis, as well as to provide an accurate prognosis. At present, mostly antibodies are used in clinical applications and cancer diagnostics, such as immunohistochemistry, flow cytometry, the detection of tumor markers, in vivo imaging, etc. [80,81,82]. However, antibodies come with certain disadvantages, such as immunogenicity, poor stability, and high production costs due to limited and time-consuming production methods, which limit their usefulness to a certain extent. While aptamers can bind targets with high affinity and specificity comparable to antibodies, aptamers have obvious advantages when it comes to stability, chemical modifications, and production cost [83]. Aptamers have therefore found vast applications in the field of cancer diagnosis, such as immunohistochemical analysis, in vivo imaging, and the detection of CTCs.

In addition, numerous aptamers against specific cancer biomarkers have been utilized for the capture and analysis of targets from liquid biopsy specimens, including aptamers against PSMA, prostate-specific antigen (PSA), mucin 1 (MUC1), and epidermal growth factor receptor 2 (HER2) [84,85,86]. Molecular markers on the surface of exosomes, such as CD63, and the extracellular domain of HSP70 have been targeted for the development of specific aptamers to isolate and analyze EVs [87,88]. Herein, we summarize some of the existing aptamers successfully applied in liquid biopsy and their applications for various cancer types (Table 1).

### 3.1. CTCs from Liquid Biopsy

CTCs are cells detached from solid cancer tissues circulating in the bloodstream. They are believed to be responsible for the hematogenous spread of cancer cells essential for the development of distant metastasis [120]. CTCs retain unique cellular structures and functions and can provide real-time information about the originating cancer issues. This makes them an ideal target for diagnostics as well as for guiding therapy.

Over the past years, CTC-based liquid biopsy has found applications in a variety of clinical settings including early cancer diagnosis, treatment monitoring, prognosis, and personalized cancer therapy [121]. Aptamers, with their unique properties and advantages, are an exciting new tool for the isolation and detection of CTCs.

#### 3.1.1. Capture

CTCs from different cancers have been successfully isolated with aptamer-based magnetic separation [122,123,124]. For this approach, a magnetic field is applied to separate magnetically tagged target cells from their respective pool [125]. The disadvantage of this method is the use of a single aptamer, which runs the risk of losing some CTC subpopulations due to the unequal expression of CTC biomarkers. Sheng et al. utilized gold nanoparticles (AuNPs) as scaffolds to assemble multivalent aptamer nanospheres, which were further coated into micro- channels for isolation of leukemia cells in whole blood (Figure 3A) [126]. Up to 95 aptamer ligands were attached onto each gold nanoparticle (approximately 14 nm). When the microchannel of a microfluidic chip was modified by gold nanoparticles (AuNPs), a 39-fold increase in binding affinity was developed compared to a flat surface coated with aptamer alone. The results showed that the efficiency of cell capture increased from 49% by using aptamer alone to 92% by using AuNP aptamer, indicating the strong potential of AuNP-based microfluidic chip devices for the analysis of CTCs. Zheng et al. employed barcode particles of spherical colloidal crystal clusters decorated with dendrimer-amplified aptamer probes to capture, detect, and release multiple types of CTCs (Figure 3B) [127]. This barcode-particle technology can simultaneously capture, detect, and release multiple types of CTCs from a complex sample with a high capture efficiency of 92.83%. Shen and co-workers developed a new-generation NanoVelcro chip (Figure 3C) [128]. Two aptamers were coated on silicon nanowire substrates (SiNWS), which were utilized to immobilize CTCs in a stationary device setting. Then, SiNWS was integrated with a polydimethylsiloxane (PDMS)-based chaotic mixer that enhanced the contact frequency between flow-through cancer cells and the substrate, thereby improving the capture efficiency of the CTCs. At optimized conditions, the capture efficiency exceeded 80% for A549 cells in artificial blood samples, and the release efficiency was more than 85%. This SiNW-based microfluidic chip platform can not only improve CTC capture efficiency, but can also realize controllable CTC release via nuclease treatment. In another report, Liu et al. developed a homogeneous fluorescent method to detect cancer cells based on catalytic hairpin assembly (CHA) and bifunctional aptamers (Figure 3D) [129]. The strategy showed high specificity for discriminating normal cells and leukocytes, and the detection limit was 10 cells/mL. Zamay et al. used a targeted selection of DNA aptamers and applied two different aptamer clones for the isolation and detection of CTCs in peripheral blood samples from patients with different lung cancer types and benign lung tumors [130]. In addition, these aptamers were further utilized for a bioluminescent solid-phase sandwich-type microassay to detect lung tumor elements circulating in the blood [131]. Fang’s group introduced a cocktail of aptamer-modified magnetic nanoparticles (apt-MNPs) to capture heterogeneous CTC populations [132]. A microwell chip was used to improve the CTC purity for downstream applications. Eventually, platforms were developed to simultaneously capture and analyze CTCs in situ. Another example is the magnetic nanoparticle-quantum dot (QD)-aptamer copolymers by Li et al., which isolated CTCs with about 80% efficiency and simultaneously phenotyped them using a fluorescent approach [133]. Magnetic-based affinity isolation is easy to perform, can be conveniently integrated into various platforms, and has a high capture efficiency; however, CTC purity can be a concern due to the nonspecific binding of magnetic tags to other cells in the bloodstream. In some devices, a uniform multiscale TiO2 nanorod array is fabricated to provide a “multi-scale interacting platform” for cell capture by Sun et al. [134]. The platform showed up to 85–95% capture yield of the target cancer cells on the BSA-aptamer TiO2 nanorod substrates, revealing the potential application of the TiO2 nanorods for the efficient and sensitive capture of rare CTCs.

#### 3.1.2. Release

After the successful capture of CTCs, it is critical to release them from whatever matrix they are bound to so as to allow for downstream processing and analysis. Different from other recognition ligands, aptamers can be easily manipulated to allow for the convenient release of CTCs. The aptamer-CTC bond can be disrupted either by the denaturation of the aptamer or by the detachment of the aptamer from the capture interface. Disruption of the aptamer-CTC interaction can be achieved by physical or chemical strategies such as adsorption, covalent binding, or the use of linker molecules to allow for the release of the CTCs [135].

Au−S bonds, for example, can be easily disrupted by electrochemical reduction and competitive ligand binding, which leads to the release of the CTCs bound to aptamers by Au substrates [136,137,138,139]. Yang’s group created an AP-Octopus-Chip using thiol-terminated aptamer-functionalized AuNPs for capturing CTCs. After capture, glutathione (GSH) with three thiol moieties was utilized to release the CTCs through competitive binding to AuNPs with more than 80% efficiency and 96% viability [139]. Aside from the direct disruption, different approaches have been developed to alter the aptamer-interface interaction. Wang’s group developed an elegant approach using UV light to release CTCs. They used azobenzene-tagged aptamers that interact with a cyclodextrin (CD)-modified surface. Exposure to UV light leads to a switch from trans- to cis-isomers, which then leads to an unmatching of the host-guest pairs, thereby releasing the captured CTCs [140].

#### 3.1.3. Analysis

Downstream detection and analysis have been at the core of CTC research over the past decade. Clinical studies have suggested a correlation between the detection and quantification of CTCs and clinical outcomes, proposing CTCs not only as a diagnostic, but also as a prognostic tool. The opportunity to monitor treatment response via the detection and quantification of CTCs can be used to directly and efficiently guide treatment strategies. As CTCs are believed to be the origin of distant metastasis, the elimination of CTCs might even become useful as an additional cancer therapy option, potentially preventing metastatic cancer spread.

Obtaining CTCs from blood samples is far less invasive than obtaining a surgical specimen or even performing an interventional biopsy. Nonetheless, it provides vast genotypic and phenotypic information on the original cancer, potentially replacing histologic confirmation—which is generally required before initiating cancer treatment—in the future. In addition, CTCs are easy to analyze and manipulate in the laboratory, making for an ideal research platform to gain insights into the mechanism of tumorigenesis and cancer progression, as well as guiding personalized cancer therapy.

Fluorescence assays are widely used for the identification and detection of CTCs. They are highly sensitive, easy to manipulate, and can be conveniently combined with various different isolation methods. However, autofluorescence background is a major issue, interfering with the sensitive and reliable detection of CTCs [129,141,142,143,144]. To overcome this disadvantage, Liu’s group labeled tumor cells with aptamer-conjugated upconversion nanoparticles (UCNPs). In a second step, magnetic nanoparticles were used to further isolate the labeled tumor cells. The resulting upconversion luminescence imaging was free of autofluorescence and allowed for the sensitive and specific detection of CTCs in blood samples [129]. Near-infrared (NIR) fluorescent probes can also be useful by offering reduced light scattering and minimal autofluorescence from living tissues when detecting CTCs from the complex blood matrix. Ding et al. were able to detect as little as 5 CTCs per mL of sample using NIR fluorescent nanoprobes based on aptamer-modified Ag2S nanodots [143]. The turn-on method is another strategy to work around fluorescence interference. Fluorescence detection is limited by the performance of fluorescent tags, including fluorescence intensity, labeling specificity, and stability, among others. The refinement and optimization of fluorescent tags will be needed to further improve the sensitive and reliable detection of CTCs.

### 3.2. EVs from Liquid Biopsy

EVs are lipid bilayer-delimited particles that are secreted from most cell types and can be found in all biological fluids, such as blood, saliva, pleura fluid, and urine [87,145,146]. EVs contain a unique set of proteins, DNA, and RNA from the originating cells and act as mediators of intercellular communication [147]. EVs are grouped by size and origin into exosomes (30−200 nm), microvesicles (200−1000 nm), and apoptotic bodies (>1000 nm) [148]. Tumor-derived EVs carry information about cancer cells and are believed to be important players in cancer progression and metastasis, as well as the development of drug resistance. Tumor-derived EVs are therefore promising biomarkers not only for the early detection of cancer, but also for the evaluation of therapeutic response, the assessment of prognosis, and the monitoring of cancer recurrence and/or metastasis [87,146]. Despite the exciting potential, there are still many technical challenges to overcome with regard to the isolation and detection of EVs [149]. Currently, size, density, surface charge, membrane proteins, and lipid membranes have been explored for the isolation of EVs from various body fluids. Among these approaches, bioaffinity-based EV isolation targeting unique EV membrane antigens such as EpCAM, EGFR, HER2, PSA and Tim4 requires specific capture molecules with high affinity [150,151,152]. Aptamer-based detection of EVs has emerged as an advantageous strategy for the efficient and specific isolation of EVs utilizing affinity-based isolation and quantitative detection using signal transduction, as well as signal amplification.

#### 3.2.1. Capture

EVs can be isolated utilizing antibodies or aptamers as recognition ligands. The advantage of aptamer-based isolation is the fast and efficient release of EVs, low immunogenicity, and an easy incorporation of downstream processing such as signal transformation and amplification for the further analysis of EVs. Zhang et al. could isolate EVs in as little as 1.5 h with an efficiency of ~78% (Figure 4A) [97]. By using complementary sequences to release the EVs, the isolated EVs remained stable and functional, allowing for further analysis. Aptamer-anchored DNA nanostructures can be designed according to specific substrates for capture and detection, and can easily be controlled and manipulated. The more efficient and thorough the capture and isolation of the EVs, the better the conditions for the downstream analysis, biofunctional studies, and biomedical applications of the EVs. Dong et al. used aptamer magnetic bead bioconjugates to capture tumor exosomes (Figure 4B) [153]. A strategy based on aptamer recognition-induced multi-DNA release was implemented, where the recognition of the exosomes causes the release of the detected complementary strands and the recognition event produces a decrease in the voltammetric signal of the marker in response to the exosome target.

Xue et al. reported a multivalent, long single-stranded aptamer with repeated units for EV enrichment and retrieval [154]. EVs were secured by biotin-labeled multivalent aptamers (MAs), and 45% of EVs were isolated from the spiked samples in 40 min with a depletion of 84.7% of albumin contamination. Furthermore, 93.1% of the isolated EVs were retrieved via DNase-mediated aptamer degradation in 10 min for downstream molecular analyses.

#### 3.2.2. Detection and Quantification of EVs

The detection and quantification of cancer-derived EVs can aid in early cancer diagnosis and the assessment of therapeutic response. A large variety of approaches and technologies utilizing novel aptamer-based methods have been implemented to detect and quantify cancer-derived EVs in complex biological or clinical samples.

Kelly’s group used an elegant electrochemical approach, wherein they constructed an on-chip multiplexed electrochemical sensor and used aptamer-modified metal nanoparticles as probes to detect and quantify EVs [155]. Zhou et al. developed an aptamer-based electrochemical biosensor, which could detect as few as 1 × 10^6^ particles/mL of exosomes by integrating Au electrode arrays in microchannels [156]. EVs were targeted via aptamers specific to transmembrane protein CD63, which were also hybridized to redox-labeled probing chains. Once bound, conformational change resulted in the release of redox reporters and, thus, in a decreased electrochemical signal. A nanotetrahedron (NTH)-assisted aptasensor was developed by Tan’s group. They linked DNA-based nanostructure and portable electrochemical devices to directly capture and detect hepatocellular exosomes. The NTH-assisted aptasensor was 100 times more sensitive for the detection of EVs than the single-stranded aptamer-functionalized aptasensor (Figure 4C) [92]. While electrochemical detection comes with excellent sensitivity, its practical application is limited by the rather low stability of EVs within body fluids. Another easy and convenient method for detecting EVs is a colorimetric approach. Colorimetric tools can be used to quantify and profile exosomal protein information in a simple manner. Aptamers have the ability to enhance nanozymes, nanoparticles with peroxidase-like activity, which can be used for colorimetric experiments [85,157,158]. Wang et al. attached aptamers to graphitic carbon nitride nanosheets (g-C3N4 NSs) [158], which detached from the sheets once bound to EVs. This resulted in a decrease in the peroxidase activity of the nanozymes, leading to a color change, which could be quantified using spectrometry. A third detection strategy utilizes fluorescence. Target proteins on the surface of EVs are labeled with small aptamers (2–3 mm) without significantly changing the overall size. Sun’s group used two kinds of fluorescent dye-labeled aptamers to label EVs. This was followed by on-chip λ-DNA-mediated separation by size (Figure 4D) [91]. Fluorescent analysis demonstrated a heterogeneous marker expression even within single EV subpopulations. This approach was further advanced by the same group to profile EV surface proteins via a size-dependent thermophoretic strategy. With a thermophoretic aptasensor (TAS) using a panel of seven fluorescent aptamers to profile EV surface proteins and then separating them by size, they were able to classify cancer types with an accuracy of 68%, a sensitivity of 95%, and a specificity of 100% [148].

However, especially in early stage cancer, the efficient detection and quantification of EVs with low protein expression levels remains a challenge. Several signal amplification strategies, including direct and competitive amplifications, have been utilized. For direct amplification, aptamers bind to EVs to directly trigger signal amplification reactions, such as rolling circle amplification (RCA) or hybridization chain reaction (HCR) [93,159,160]. In the competitive amplification assay, aptamers are released from their carrier once bound to EVs, thus starting a signal amplification reaction [151].

#### 3.2.3. Analysis

EV secretion has been found to be increased in response to various stimuli, such as inflammation, hypoxia, and acidic microenvironments, among others. In cancer, EV secretion levels have been shown to correlate with the invasiveness of the cancer [161,162]. The heterogeneity of EV subpopulations can be accurately captured by the multiplex analysis of EV biomarkers, which can be performed at a single EV level [147,163,164]. Lee et al. used this approach to characterize single EVs by multiplex protein analysis with eleven markers. For this, EVs were first immobilized in a microfluidic chamber, then fluorescently stained in several rounds and analyzed [163]. Lu’s group used a multiplex barcode technique to profile single-cell EV secretion from thousands of human oral squamous cell cancer cells [147]. They demonstrated an association between the downregulation of certain EV phenotypes and the aggressiveness of the originating cancer cell. Apart from protein analysis approaches, RNA analysis approaches of EVs have been developed to facilitate the detection of early stage cancer and to screen for drug resistance [165,166]. Weissleder’s group, for example, identified mRNA markers associated with drug resistance using a microfluidic chip. This chip combined immunomagnetic capture, on chip RNA isolation, and the real-time RNA analysis of EVs [166]. Furthermore, the detection of DNA mutations and microRNAs in EVs was found to facilitate cancer diagnosis [167,168]. Various miRNAs, as well as associated proteins, have been identified within cancer derived EVs which have the potential to be used as biomarkers, such as CPNE3 in colorectal cancer EVs [169], miR-451a in non-small cell lung cancer EVs [170], and miR-451a in pancreatic ductal adenocarcinoma EVs [171].

## 4. Conclusions and Future Perspective

In this review, we discussed recent advances in aptamer-mediated liquid biopsy with a focus on CTC and EVs, their isolation, release, and analysis. Clinical application of liquid biopsy is already paving the way for precision medicine, and the main advantage of the liquid biopsy approach stands with the possibility of capturing tumor heterogeneity as a whole. Liquid biopsy is an easy, minimally invasive, and reproducible way to obtain CTCs and cancer-derived EVs, which are powerful platforms for cancer diagnosis; the real-time monitoring of treatment response; and, ultimately, personalized cancer therapy. Aptamers themselves are an ideal tool for the isolation and detection of CTCs and EVs. They are small in size; fast and cheap to synthesize; can be easily modified and integrated in downstream analysis and various applications; and—most importantly—detect CTCs and EVs with high efficiency, sensitivity, and specificity.

Despite their clinical potential, there are certain challenges and limitations of aptamer-based liquid biopsies, and their excellent features and versatility need to be reinforced with reliable clinical applications. Aptamers may behave differently within the context of complex body fluids. They can become degraded by nuclease activity and suffer decreased affinity compared to their in vitro function. Most reports of aptamer-based liquid biopsy approaches lack information on clinical standards, such as sampling, preservation, the processing of samples, detection, and readout, limiting the actual clinical utility at this time. To reach the goal of clinical applications of aptamer-based liquid biopsy, clinical standards, reagent kits, and standard instruments have to be developed to streamline the process and allow them for use as part of the clinical routine. In addition, new selection and modification strategies have to be developed to generate reliably and highly performing aptamers.

In comparison to antibodies, aptamer-based technologies are still at an early stage of development, and further research is needed to ensure their widespread adoption. Recent advances in next-generation sequencing (NGS) and the use of high-throughput NGS can be expected to have a huge impact on the aptamer field. Interestingly, therapeutic applications of aptamers have been slow to catch on, despite many clinical trials. The U.S. Food and Drug Administration (FDA) has, so far, only approved Pegaptanib (Macugen), a 28-base ribonucleic acid aptamer, against all isoforms of human vascular endothelial growth factor (VEGF) for clinical use [172]. However, with the recent expiration of the patent on SELEX technology, we can expect more aptamer-based drugs in the near future. Due to their unique three-dimensional structure, aptamers can bind targets with high affinity and specificity. They can be easily coupled to various platforms due to their small size and ease of manipulation. Combining these advantages, aptamer technology has the potential for innumerable clinical applications, from cancer detection and therapy to the treatment of bacterial or viral infections and targeted drug delivery. Aptamer-based approaches can offer an extremely high resolution and will therefore be an invaluable tool for shaping the future landscape of personalized medicine.

## Figures and Tables

**Figure 1 ijms-22-05601-f001:**
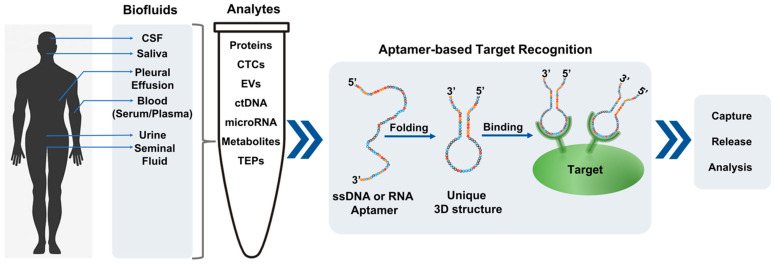
Schematic diagram for common aptamer selection and their applications for circulating tumor targets in liquid biopsy. CSF: Cerebrospinal fluids; CTCs: Circulating tumor cells; ctDNA: Circulating tumor DNA; TEPs: Tumor-educated platelets; EVs: Extracellular vesicles; ssDNA: Single-stranded DNA.

**Figure 2 ijms-22-05601-f002:**
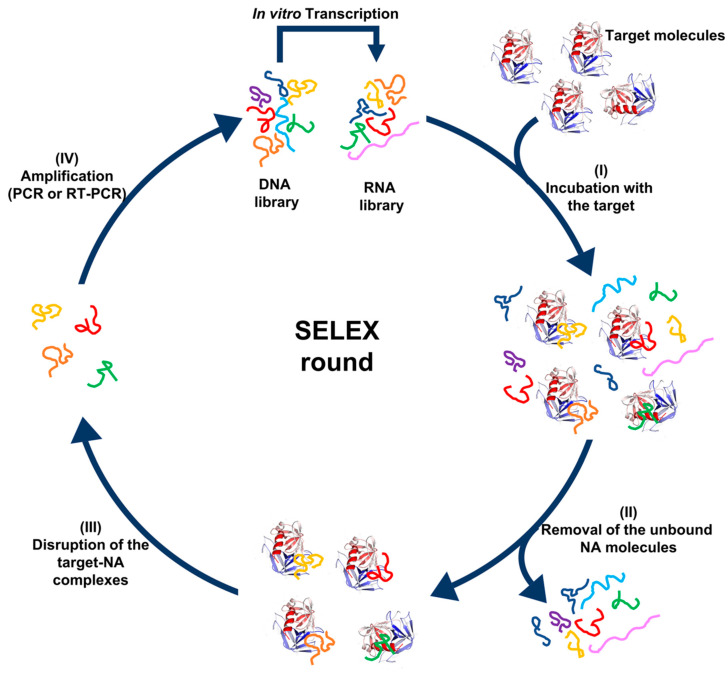
Schematic diagram of the SELEX method using DNA and RNA libraries.

**Figure 3 ijms-22-05601-f003:**
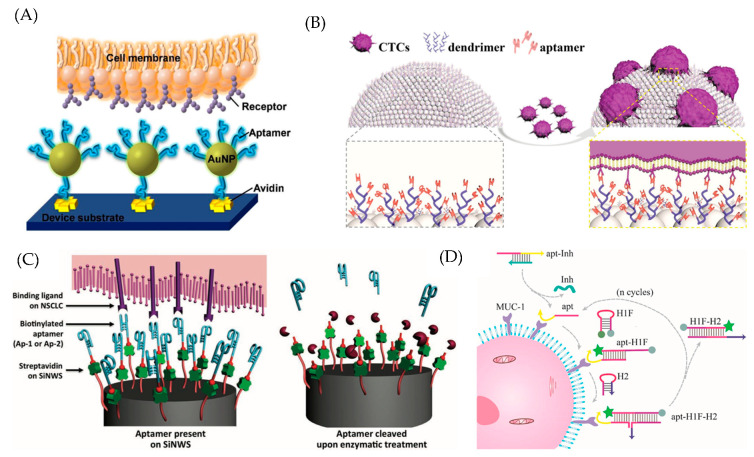
Schematic illustration of aptamer-based techniques for the capture and detection of circulating tumor cells (CTCs). (**A**) Multivalent AuNP-aptamer for the enhanced capture of tumor cells on microdevices. Reprinted with permission from ref. [126]. (**B**) SiNW-based platform for CTC capture and release with temperature stimulation. Reprinted with permission from ref. [127]. (**C**) NanoVelcro Chip consisted of aptamer-coated silicon nanowire substrate (SiNWS) and an overlaid PDMS chaotic mixer. Reprinted with permission from ref. [128]. (**D**) Bi-functional aptamer-mediated catalytic hairpin assembly for the fluorescence turn-on detection of rare cancer cells. Reproduced with permission from ref. [129].

**Figure 4 ijms-22-05601-f004:**
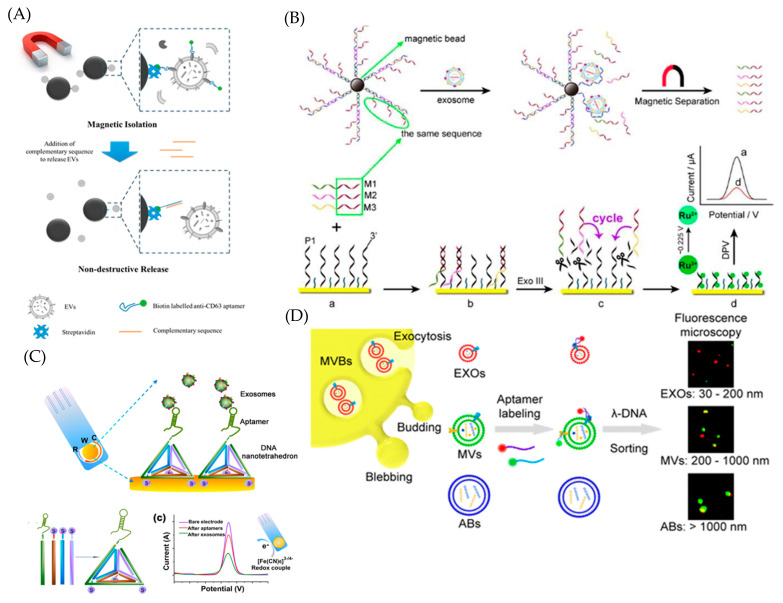
Schematic illustration of aptamer-based techniques for the capture and detection of EVs. (**A**) DNA aptamer-based magnetic isolation system (AMI) for the rapid capture and nondestructive release of EVs. Reprinted with permission from ref. [97]. (**B**) Electrochemical detection of tumor exosomes based on aptamer recognition-induced multi-DNA release. Reprinted with permission from ref. [153]. (**C**) Nanotetrahedron-assisted aptasensor for the electrochemical detection of cancerous exosomes. Reproduced with permission from ref. [92]. (**D**) λ-DNA-mediated sorting of EV subpopulations and aptamer-mediated fluorescence analysis of EVs. Reproduced with permission from ref. [91].

**Table 1 ijms-22-05601-t001:** A List of Selected Aptamers Used in Liquid Biopsy by Cancer Types.

Aptamer	Tumor Type	Application
A8	Breast, lung, and ovarian cancer	Quantification of tumor-derived exosomes [89]
MO-1, MO-2	Cervical cancer	Detection of cancer-specific EVs [90]
H2 and SYL3C	Breast cancer	Detection of cancer-specific EVs [91]
LZH8	Hepatocellular carcinoma	Detection of cancer-derived exosomes [92]
MUC_3	Gastric cancer	Detection of cancer-derived exosomes [93]
EpCAM/Ep114	Breast cancer, colorectal cancer	Detection of cancer-specific EVs [94,95]
CD63	Gastric, lung, and breast cancer	Detection and quantification of cancer-derived EVs [94,96,97,98]
H2, CEA, and PSMA	Breast, colorectal, and prostate cancer	Detection of cancer-derived exosomes [99]
PL1-8	Pancreatic ductal adenocarcinoma	Biomarkers identification, drug delivery [100]
Aptamers 1 and 146	Pancreatic ductal adenocarcinoma	Detection of CTC [101]
AS1411	Breast cancer	Drug-targeting nanovesicles [102]
PSMA and EGFR	Breast, colorectal, and prostate cancer	siRNA delivery via aptamer-functionalized EVs [103]
sgc8	Leukemia, lymphoma	Aptamer-functionalized exosomes for drug delivery [104]
CTLA-4	Melanoma, lymphoma, colon cancer, renal cell cancer, fibrosarcoma	Immune-checkpoint blockade [105]; targeting STAT3 siRNA [106]
PD1	Colon cancer	Immune-checkpoint blockade [107]
TIM3	Colon cancer	Immune-checkpoint blockade [108]
IL10R	Colon cancer	Immune-checkpoint blockade [109]
IL6	Glioma and hepatoma	In vitro growth inhibition [110]
IL4R	Breast cancer	Targeting myeloid-derived suppressor cells (MDSC) and tumor-associated macrophages (TAM) [111]
4-1BB	Mastocytoma, melanoma, colon cancer, breast cancer, oncogene-induced high-grade glioma, MCA fibrosarcomas	Costimulatory receptor agonist [112]; targeting costimulation to the tumor [113,114]
OX40	Melanoma	Costimulatory receptor agonist [115]
CD28	Lymphoma, melanoma	Costimulatory receptor agonist [116]; targeting costimulation to the tumor [53]
CD40	Lymphoma	Stimulatory receptor agonist [117]
CD16α	Lysis of human gastric and lung cancer cell lines in vitro	Antibody-dependent cell-mediated cytotoxicity (ADCC) [118]
BAFF-R	Mantle cell lymphoma	Targeted STAT-3 inhibition [119]

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
