# Peer review of "The Potential of Aptamer-Mediated Liquid Biopsy for Early Detection of Cancer"

_ijms, 2021, doi:10.3390/ijms22115601_

Round 1

Reviewer 1 Report

In the review The potential of aptamer-mediated liquid biopsy for early detection of cancer” D. Royand and coauthors analyzed currant data on the aptamers (of both types, DNA and RNA) selection and application in medical diagnostics, partially, in oncology. Nowadays the aptamer-based analytical technology are developing extremely actively, and the analyses, presented in this review helps not to miss something the most important and interesting. The advantages of aptamers over antibodies as molecules for target recognition and binding are attracting the attention of an increasing number of specialists. Note, that the bibliography includes 166 titles, more then 70% of which were published in the last 5 years, so it really reflects the state of the art in research in this area. It was a good idea to focus on the data regarding detection of the circulating tumor cells and extracellular vesicles as the most informative and accessible cancer-related targets. But on the Reviewer opinion there is not enough information concerning aptamer-based analysis of lung cancer, one of the most common and dangerous. It may be useful here to pay attention to a series of works by Canadian and Russian researchers (see, e.g., Zamay GS et al., 2019, doi: 10.3390/cancers11030351).

Speaking about the detection of the aptamer-target complex, the authors discussed the most commonly used fluorescence assay, highlighting its drawback - a high background level and approaches to overcome it. In addition, there are photolabels that do not have this drawback - bioluminescent proteins (luciferases and photoproteins), with a simple and fast reaction and a high quantum yield, ensuring high sensitivity of the detection (see e.g., Bashmakova et al., 2019, https://doi.org/10.1016/j.talanta.2019.03.030).

In general, I can conclude that the review is well organized, clearly presented, is interesting and important. On my opinion, the paper can be published after minor revision.

Some small remarks:

  1. There is no reference to Figure 2 in the text

  2. The Reference list should be corrected in accordance with the Journal rules

Author Response

Replies to the Reviewers: 

Reviewer: 1 

Comments:

In the review “The potential of aptamer-mediated liquid biopsy for early detection of cancer” D. Roy and and coauthors analyzed currant data on the aptamers (of both types, DNA and RNA) selection and application in medical diagnostics, partially, in oncology. Nowadays the aptamer-based analytical technology are developing extremely actively, and the analyses, presented in this review helps not to miss something the most important and interesting. The advantages of aptamers over antibodies as molecules for target recognition and binding are attracting the attention of an increasing number of specialists. Note, that the bibliography includes 166 titles, more then 70% of which were published in the last 5 years, so it really reflects the state of the art in research in this area. It was a good idea to focus on the data regarding detection of the circulating tumor cells and extracellular vesicles as the most informative and accessible cancer-related targets. But on the Reviewer opinion there is not enough information concerning aptamer-based analysis of lung cancer, one of the most common and dangerous. It may be useful here to pay attention to a series of works by Canadian and Russian researchers (see, e.g., Zamay GS et al., 2019, doi: 10.3390/cancers11030351).

Speaking about the detection of the aptamer-target complex, the authors discussed the most commonly used fluorescence assay, highlighting its drawback - a high background level and approaches to overcome it. In addition, there are photolabels that do not have this drawback - bioluminescent proteins (luciferases and photoproteins), with a simple and fast reaction and a high quantum yield, ensuring high sensitivity of the detection (see e.g., Bashmakova et al., 2019, https://doi.org/10.1016/j.talanta.2019.03.030).

In general, I can conclude that the review is well organized, clearly presented, is interesting and important. On my opinion, the paper can be published after minor revision.

Ans: We appreciate your valuable comments. According to your suggestion, additional sentences and references have been incorporated as appropriate to address your comments in the revised manuscript.

In the revised manuscript:

Page 9, line 223: Zamay et al. used a targeted selection of DNA aptamers and applied two different aptamer clones for isolation and detection of CTCs in peripheral blood samples of patients with different lung cancer types and benign lung tumors [126]. In addition, these aptamers were further utilized for a bioluminescent solid-phase sandwich-type microassay to detect lung tumor elements circulating in blood [127].  ...

References

  1. Zamay, G.S.; Kolovskaya, O.S.; Ivanchenko, T.I.; Zamay, T.N.; Veprintsev, D.V.; Grigorieva, V.L.; Garanzha, I.I.; Krat, A.V.; Glazyrin, Y.E.; Gargaun, A. Development of DNA aptamers to native EpCAM for isolation of lung circulating tumor cells from human blood. Cancers 2019, 11, 351.
  2. Bashmakova, E.E.; Krasitskaya, V.V.; Zamay, G.S.; Zamay, T.N.; Frank, L.A. Bioluminescent aptamer-based solid-phase microassay to detect lung tumor cells in plasma. Talanta 2019, 199, 674-678.

Some small remarks:

There is no reference to Figure 2 in the text

The Reference list should be corrected in accordance with the Journal rules

Ans: Thank you. We have mentioned Fig. 2 in the text and revised the reference list according to the journal’s format.

In the revised manuscript:

Page 2, line 64: ….characterized (Fig. 2).

Reviewer 2 Report

In this review, the authors summarize recent progress in aptamer-based liquid biopsy for cancer detection, with a focus on the isolation and detection of circulating tumor cells (CTCs) and extracellular vesicles (EVs). Finally, future perspectives

But the focus has been extensively described and discussed in this other review:

Liquid Biopsy: Application in Early Diagnosis and Monitoring of Cancer

Junjie Feng  Bo Li  Jiaxu Ying  Weilun Pan  Chunchen Liu  Tingting Luo  Huixian Lin  Lei Zheng

First published: 27 September 2020 https://doi.org/10.1002/sstr.202000063

Small Structures Volume1, Issue3 December 2020

Comparing the two reviews are similar, therefore it is not suitable for publication.

Author Response

Reviewer: 2

Comments:

In this review, the authors summarize recent progress in aptamer-based liquid biopsy for cancer detection, with a focus on the isolation and detection of circulating tumor cells (CTCs) and extracellular vesicles (EVs). Finally, future perspectives

But the focus has been extensively described and discussed in this other review:

Liquid Biopsy: Application in Early Diagnosis and Monitoring of Cancer

Junjie Feng  Bo Li  Jiaxu Ying  Weilun Pan  Chunchen Liu  Tingting Luo  Huixian Lin  Lei Zheng

First published: 27 September 2020 https://doi.org/10.1002/sstr.202000063

Small Structures Volume1, Issue3 December 2020

Comparing the two reviews are similar, therefore it is not suitable for publication..

Ans: Thank you for your comment. The application of liquid biopsies is becoming more common for cancer diagnosis, prognosis, and personalized therapy. In this review, we specifically address aptamers and their comprehensive applications in liquid biopsy as also mentioned by other reviewers. We also outline some of the analytical challenges encountered using liquid biopsy techniques and focus on the isolation and detection of circulating tumor cells (CTCs) and extracellular vesicles (EVs) by various aptamer approaches. However, to expand on the general applications of liquid biopsy in cancer management we have added your suggested reference.

References

  1. Feng, J.; Li, B.; Ying, J.; Pan, W.; Liu, C.; Luo, T.; Lin, H.; Zheng, L. Liquid Biopsy: Application in Early Diagnosis and Monitoring of Cancer. Small Structures 2020, 1, 2000063.